# Design and Experiment of Plate Taking Control System of Edible Sunflower (*Edulis helianthus catino* L.) Harvester

Feng Pan [1,2], Jincheng Chen [1,2], Hui Zhang [1,2], Lin Han [1,2,3], Yuncheng Dong [1,2], Bin Li [1,2,*] and Chao Ji [1,2,*]

1 Institute of Mechanical Equipment, Xinjiang Academy of Agricultural and Reclamation Science, Shihezi 832000, China; 17609922366@163.com (F.P.); shznkycjc@163.com (J.C.); zhanghxj2021@163.com (H.Z.); han_lin1@163.com (L.H.); dongych2024@163.com (Y.D.)
2 Key Laboratory of Northwest Agricultural Equipment, Ministry of Agriculture and Rural Affairs, Shihezi 832003, China
3 College of Mechanical and Electrical Engineering, Shihezi University, Shihezi 832003, China
* Correspondence: bin175337620@shzu.edu.cn (B.L.); jicobear@163.com (C.J.)

**Abstract:** This study aims at the problems of high labor intensity, high cost and high loss rate of mechanical picking of seeds, low mechanization level and difficulty to guarantee the quality of picking plates in the process of picking edible sunflower. Based on the principle of manual plate taking, a plate taking control system for a sunflower harvesting table was designed. The principle of taking the plate of the edible sunflower harvesting table was analyzed. According to the actual operation requirements, the overall scheme of the sunflower plate control system is determined, and the control strategy of imitating artificial low-loss harvesting is designed. To reduce the grain loss in the process of taking the plate and improve the control accuracy of the system on the movement speed and displacement of the key components in the process of taking the plate, a trapezoidal acceleration and deceleration control algorithm is designed as the control algorithm of taking the plate. The working performance of the plate control system was verified with the absolute error, relative error and total loss rate of the harvest as objectives. Bench and field verification experiments were both carried out. The bench experiment showed that the speed error of the plate parts was not more than 0.028 m/s. In the bench experiments of the device, the maximum positioning error was 1.25 mm, the average relative error was only 0.94% and the grain loss rate was not more than 2.26%. Its result showed that the system algorithm was reliable, the positioning accuracy was high and the plate taking operation can be completed well. The field verification experiment showed that the forward speed of the unit was in the range of 0.4~0.8 m/s, and the total loss rate of harvest was less than 5%. When the forward speed is 0.6 m/s, the minimum harvest loss rate is 2.32%, which indicated the control system meets the requirements of sunflower harvesting operation.

**Keywords:** edible sunflower; harvester; plate taking control system; trapezoidal acceleration and deceleration control algorithm; field experiment

## 1. Introduction

Edible sunflower (*Edulis helianthus* L.) is mostly used for snacks and food additives [1–3]. It is an important economic crop for farmers to increase their income. It is suitable for planting in saline-alkali land and barren land. It has the characteristics of drought tolerance, salt tolerance and strong adaptability [3,4] and is mainly distributed in Xinjiang, Inner Mongolia, Shandong and other regions [5]. After maturity, the moisture content of edible sunflower is high, and it is not suitable for direct harvesting. At the same time, due to the limited drying of sites, edible sunflower is usually harvested in stages [6]. When harvesting, the sunflower rod is cut and pointed at the middle position, and then the sunflower rod (*Edulis helianthus catino* L.) is inserted into the sunflower plate, manually, to dry for 5–7 days [5–9] to reduce its moisture content. After drying, the field harvesting operation is carried out. At present, most of the harvesting sunflower plate

operations are carried out manually which has the shortcoming of high labor intensity, high cost and low efficiency. It is urgent to develop a harvesting device suitable for low-loss harvesting of edible sunflowers. At the same time, it is of great significance to reduce the labor intensity of farmers, improve operation efficiency, reduce the production cost of edible sunflowers and improve production efficiency. Due to the variety and climate of sunflower (edible sunflower, oil sunflower) in other countries, edible sunflower is mostly harvested by joint harvesting, and the cutting table is commonly used for edible sunflower and oil sunflower [10–12]. For example, the Helianthus 12000 series sunflower header developed by the company Capello in Italy (Capello Srl, Via Valle Po, 10012100 Cuneo, Italy), the Falcon series sunflower header developed by the company Rostselmash in Russia (Rostselmash, Rostov Nadon, Rostov Oblast, Russia), the G3 series sunflower header developed by the company Fantini in Italy and the ZIECLER series sunflower header developed by the company Dragotec in Germany (Dragotec International GmbH, Hub 7, 84329 Wurmannsquick, Germany.) are based on the principle of a corn header [13–16]. At present, the research on the harvesting of sunflower is still in its infancy in China. Researchers have used the principle of wheat and corn harvesting to develop a mechanical device for the harvesting of sunflower [17]. Zhang Shuangxia et al. [18] designed a sunflower harvesting header, which removed the sunflower plate from the sunflower rod by the reel. Because the water content of the sunflower plate after the reel was dried was low, the impact of the reel was large during the reel taking process, and the reel was in contact with the surface of the sunflower plate. Friction occurs, which easily causes grain shedding and high loss rate. The sunflower cutting table device designed by Zhang Yu et al. [18] cuts the sunflower rod through the reeling chain and the reeling teeth. In the process of cutting the sunflower rod, the cutting jitter will cause the sunfl ower plate to fall and the grain to be lost. The 4ZXRKS-4 type sunflower harvester was developed by Hongchang Machinery Manufacturing Co., Ltd. (Inner Mongolia Hongchang machinery manufacturing company, Toktor County, Hohhot City, Inner Mongolia Autonomous Region, China) [19,20]. The sunflower stem is introduced along the cutting table and fed by the dial chain. At the same time, the roller pulls the stem downward, and the sunflower stem is cut by the cutting knife. The sunflower plant is forced to feed by the dial chain. In the process of pulling the roller down, the sunflower plate will be greatly impacted, resulting in a large amount of grain loss.

The aim of this study was to solve the problem of grain loss in the process of plate picking operation of existing sunflower harvesting machinery in the mechanized harvesting process of edible sunflower in the Xinjiang area. In this paper, based on the principle of artificial take-up, a take-up control system for edible sunflower harvesting was designed which aimed to reduce the loss of grains in the process of take-up and provide a new idea for the development of edible sunflower harvesting machinery.

## 2. Materials and Methods

### 2.1. Overall Structure of Sunflower Harvesting Platform

The sunflower harvesting platform table is composed of a mechanical part and detection control part. The mechanical part consists of a frame, a dividing part, a conveying mechanism and a plate taking mechanism. The detection control part consists of a plug height detection sensor, a sunflower position detection sensor and a control system. The plate taking mechanism is composed of a nylon plate (with rack), synchronous belt and pulley, input shaft and gear, gear shaft and lifting gear. The structure is delineated in Figure 1.

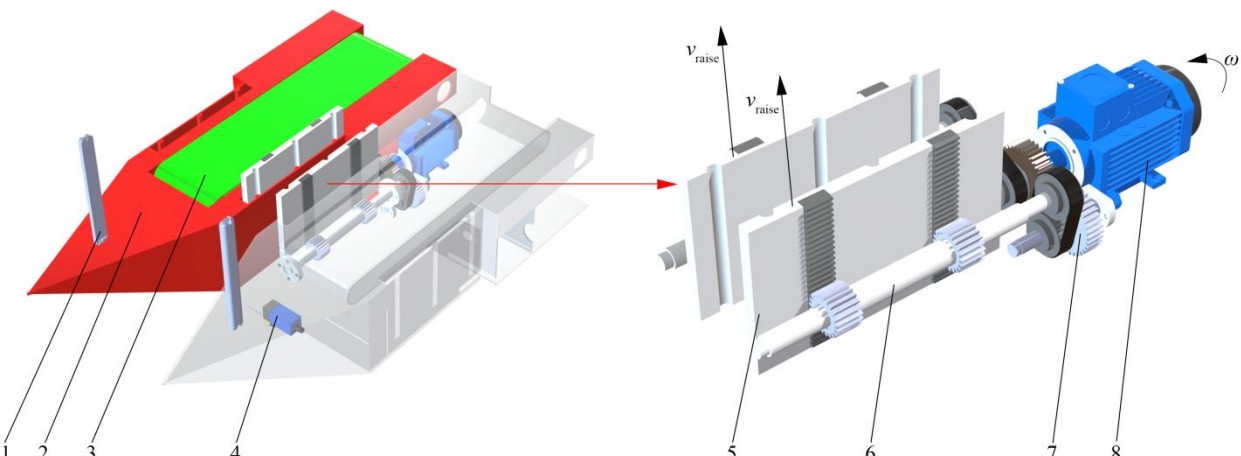

**Figure 1.** Structure diagram of edible sunflower harvesting platform. 1. Sunflower plate height detection sensor. 2. Divider. 3. Transporter. 4. Sunflower position detection sensor. 5. Nylon plate (with rack). 6. Gear shaft and lifting gear. 7. Input shafts and gears. 8. Stepper motor.

*2.2. The Working Principle of Sunflower Harvesting Platform*

When the edible sunflower harvesting platform is working, the edible sunflower plants are divided into rows by the dividing parts. Firstly, the height information of the socket is collected by the socket height detection sensor to determine the displacement of the plate taking mechanism. Then, the position information of the specific sunflower rod is obtained by the position detection sensor. According to the operation information, the plate taking mechanism is controlled to drive the nylon plate operation, and the height of the plug is profiled. The sunflower plate is taken down along the growth direction of the sunflower rod by imitating the artificial method (Figure 2), and the taken sunflower plate falls into the conveying device and is transported to the next link. The plate taking mechanism drives the nylon plate to reset to prepare for the next plate taking operation. The schematic diagram of the plate taking process is delineated in Figure 3.

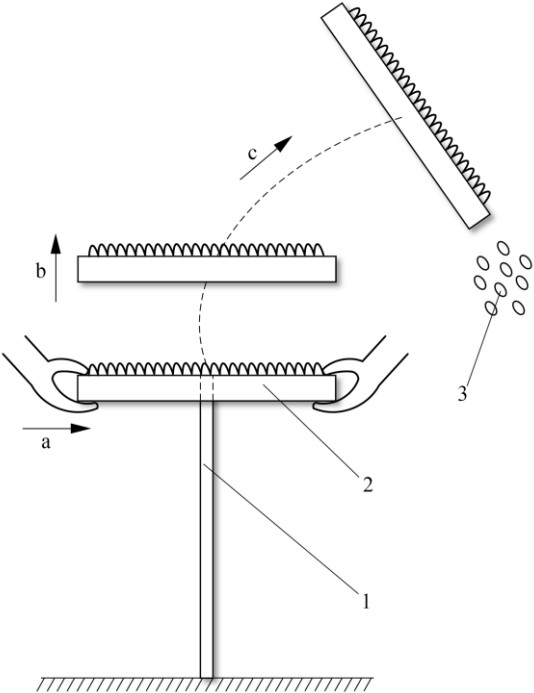

**Figure 2.** Schematic diagram of the manual pick-up. 1. Sunflower stalk. 2. Sunflower plate. 3. Sunflower seeds. a. Grasping process b. Pulling process c. Throwing process.

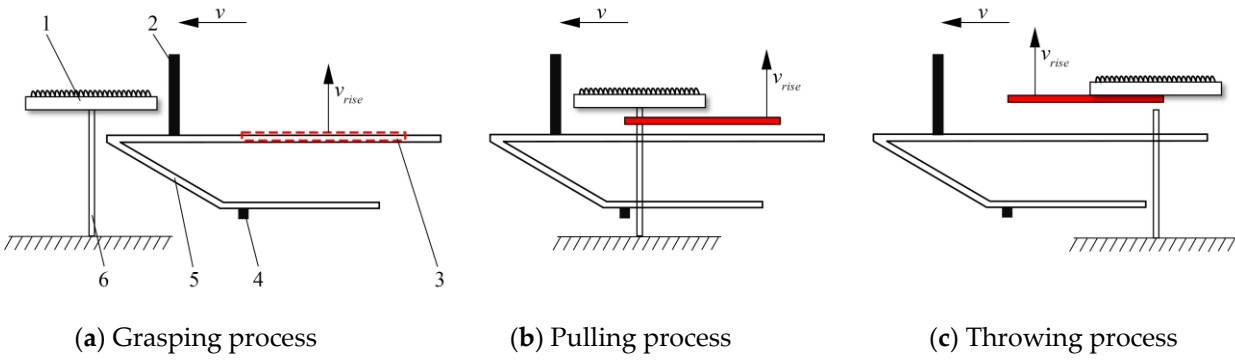

(**a**) Grasping process      (**b**) Pulling process      (**c**) Throwing process

**Figure 3.** Schematic diagram of the plate taking process. 1. Sunflower plate. 2. Sunflower plate height detection sensor. 3. Nylon plate (part for taking sunflower plate). 4. Sunflower position detection sensor. 5. Divider. 6. Sunflower stalk.

In order to ensure that the nylon plate can remove the sunflower plate before the sunflower rod is tilted, the minimum rising speed of the nylon plate should fulfill the requirement that when the sunflower rod moves to the end of the nylon plate, the nylon plate just removes the sunflower plate. The rising speed of the nylon plate can be calculated by the following Equation (1).

$$v_{rise} \geq \frac{h}{\frac{l_1}{v}} = \frac{hv}{l_1} \tag{1}$$

$$x_{rise} = h \tag{2}$$

where $v_{rise}$ is the rising speed of the nylon plate (m/s), $h$ is the height of the socket (the vertical distance from the top of the sunflower plate to the ground) (m), $v$ is the forward speed of the harvester (m/s), $l_1$ is the horizontal distance between the sunflower detection sensor and the end of the nylon plate (m), $x_{rise}$ is the rising displacement of the nylon plate (m).

After taking the plate, the nylon plate falls back to the starting point. In order to meet the conditions of single plant feeding and continuous plate taking, the nylon plate needs to fall back to the starting point when the next edible sunflower reaches the position detection sensor. The falling speed of the nylon plate can be calculated by the following Equation (3).

$$v_{fall} \geq \frac{h}{\frac{l-l_1}{v}} = \frac{hv}{l - l_1} \tag{3}$$

$$x_{fall} = h \tag{4}$$

where $v_{fall}$ is the falling speed of the nylon plate (m/s), $l$ is the plant spacing of edible sunflower plants (usually 0.6 m), $x_{fall}$ is the falling displacement of the nylon plate (m).

In order to make the picking process of sunflower harvest more stable, reduce the impact on crops during the operation of the device and reduce the loss rate, the rising and falling speed of the nylon plate should be taken as the minimum under the condition of meeting the picking speed. In order to facilitate the control, the rising and falling speeds of the nylon plate are set to be equal, and the pause time of the nylon plate rising to the target displacement and falling to the starting point is considered. The rising and falling speed of nylon plate can be calculated by the following Equation (5).

$$v_{rise} = \frac{h}{\frac{l_1}{v} - \Delta t} = v_{fall} = \frac{h}{\frac{l-l_1}{v} - \Delta t} \tag{5}$$

where $\Delta_t$ is the pause time during each pick-up process (0.025 s).

The horizontal distance between the sunflower position detection sensor and the end of the nylon plate $l_1$ can be calculated by Equation (6).

$$l_1 = \frac{l}{2} \tag{6}$$

The calculation Equations (7) and (8) of the target speed and rotation angle of the forward and reverse rotation of the stepper motor can be derived from Equations (2) and (5).

$$n_{rise} = n_{fall} = \frac{60hi}{\left(\frac{l}{2v} - \Delta t\right)\pi d} = \frac{120hvi}{(l - 2v\Delta t)\pi d} \tag{7}$$

$$\theta_{rise} = \theta_{fall} = \frac{2x_{rise}i}{d} = \frac{2x_{fall}i}{d} \tag{8}$$

In Equations (7) and (8), $n_{rise}$ is the forward target speed of the stepper motor (r/min), $n_{fall}$ is the reverse target speed for the stepper motor (r/min), $d$ is the diameter of the lifting gear (m), $i$ is the transmission ratio of the plate taking mechanism, $\theta_{rise}$ is the target rotation angle of the stepper motor forward rotation (rad), $\theta_{fall}$ is the target rotation angle of the stepper motor reverse rotation (rad).

### 2.3. Design of Plate Taking Control System

A plate taking control system based on a sunflower harvesting table was designed. The control system consists of a working speed detection sensor, a sunflower position detection sensor, a plug height detection sensor, a controller, a driver and a stepper motor, and the overall scheme of the plate taking control system is delineated in Figure 4. The control system uses the operating speed detection sensor to collect the operating speed information, the plug height detection sensor to collect the height information of the sunflower plug and the sunflower position detection sensor to detect whether the sunflower plant is in place. With the controller as the core, the controller controls the driver to drive the stepper motor, and the stepper motor drives the plate taking mechanism to realize the plate taking operation of the nylon plate.

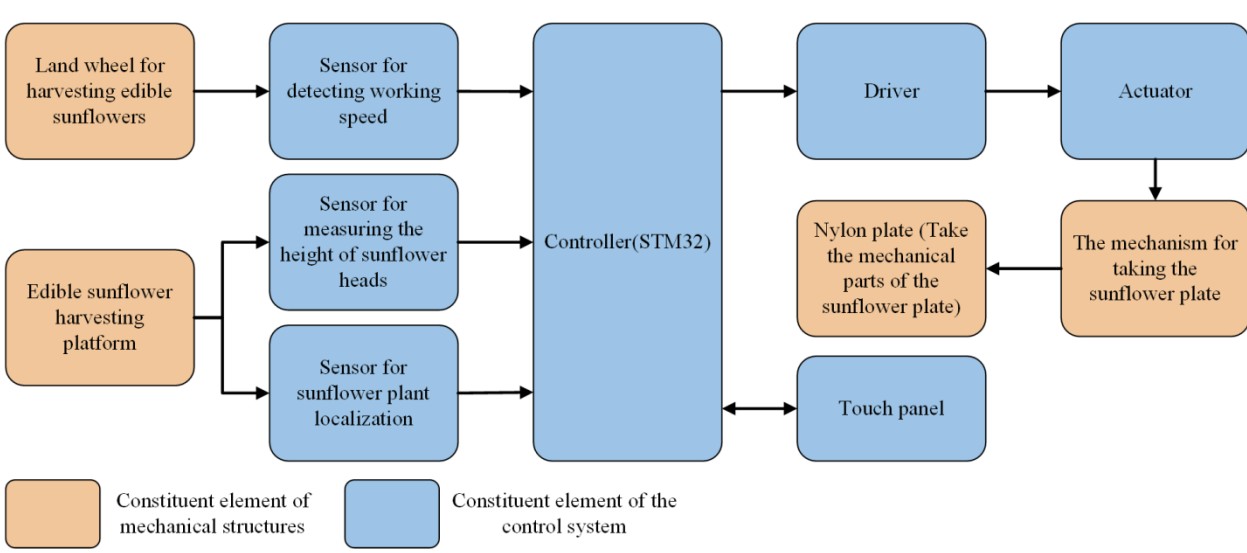

**Figure 4.** The overall structure diagram of the plate taking control system.

### 2.4. Hardware Design of Control System

The hardware of the control system is composed of a signal acquisition part, drive actuator, main control board and power supply part, as shown in Figure 5. The E6B2-CWZ3E incremental photoelectric rotary encoder was used for the position detection sensor and the working speed detection sensor of the signal acquisition part. The sunflower plate

height detection sensor was an Ip-ESCL4805 measuring light curtain. The ME-8107 travel switch is selected for the position detection sensor. The driving actuator includes an actuator and a driver. The actuator uses a Leissey 86CME120 stepper motor, and the driver uses a Leissey MA860C stepper motor driver. The main control board is composed of a controller and sensor interface circuit. The controller uses an STM32f103C8T6 Microcontroller Unit (72 main frequency, 48 pins). The sensor interface circuit includes an encoder interface circuit, measurement light curtain interface circuit and travel switch interface circuit. The interface circuit of the encoder and travel switch adopts a 5 V–3.3 V optocoupler isolation circuit. The measurement light curtain interface circuit adopts an RS485 level conversion circuit. The power supply part includes a 24 V DC power supply, a 24 V–48 V voltage conversion module and a 24 V–5 V voltage conversion module. The 24 V DC power supply is used to measure the voltage supply of the light curtain, the 24 V–48 V voltage conversion module is used to supply the voltage of the stepper motor and the 24 V–5 V voltage conversion module is used to supply the voltage of the encoder, the travel switch and the main control board.

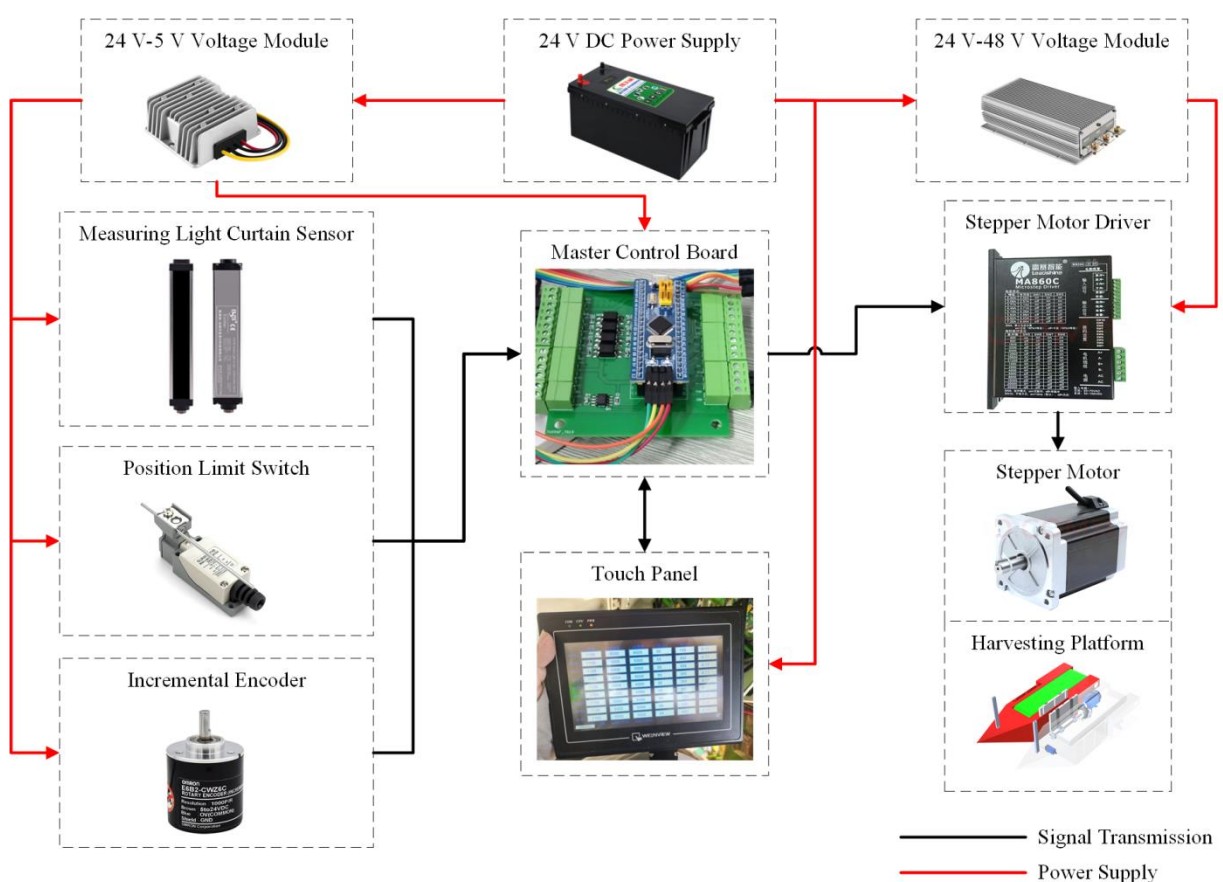

**Figure 5.** The hardware composition diagram of the control system.

## 2.5. Control System Software Design

The software flow chart of the sunflower plate taking control system is delineated in Figure 6. When the program starts to run, the system initialization is performed first. When the sunflower plant passes through the measuring light curtain, the top of the sunflower plate covers the corresponding optical axis on the measuring light curtain, and the measuring light curtain actively sends the height information of the plug plate to the controller through the RS485 ModBus-RTU protocol (the top of the sunflower plate measures the height of the bottom of the light curtain). The controller reads the measurement light curtain information through TIM4 every 20 ms. After reading the measurement light curtain information many times, the maximum sampling value is stored as the height of the plug.

At the same time, the controller reads the number of encoder pulses every 20 ms to calculate the operating speed. When the travel switch is triggered, the controller obtains the height of the plug and the working speed. According to the height of the plug and the working speed, the target displacement and the target speed of the stepper motor are calculated by the equation. The calculated target speed and target displacement of the motor are brought into the trapezoidal acceleration and deceleration control algorithm of the stepper motor, and the TIM1 output PWM signal is controlled to drive the stepper motor forward and backward. When the stepper motor is driven to rotate forward, the nylon plate is driven to rise to complete the sunflower plate harvesting operation. When the stepper motor is driven to rotate backward, the nylon plate is driven down to the starting point and prepared for the next harvesting operation.

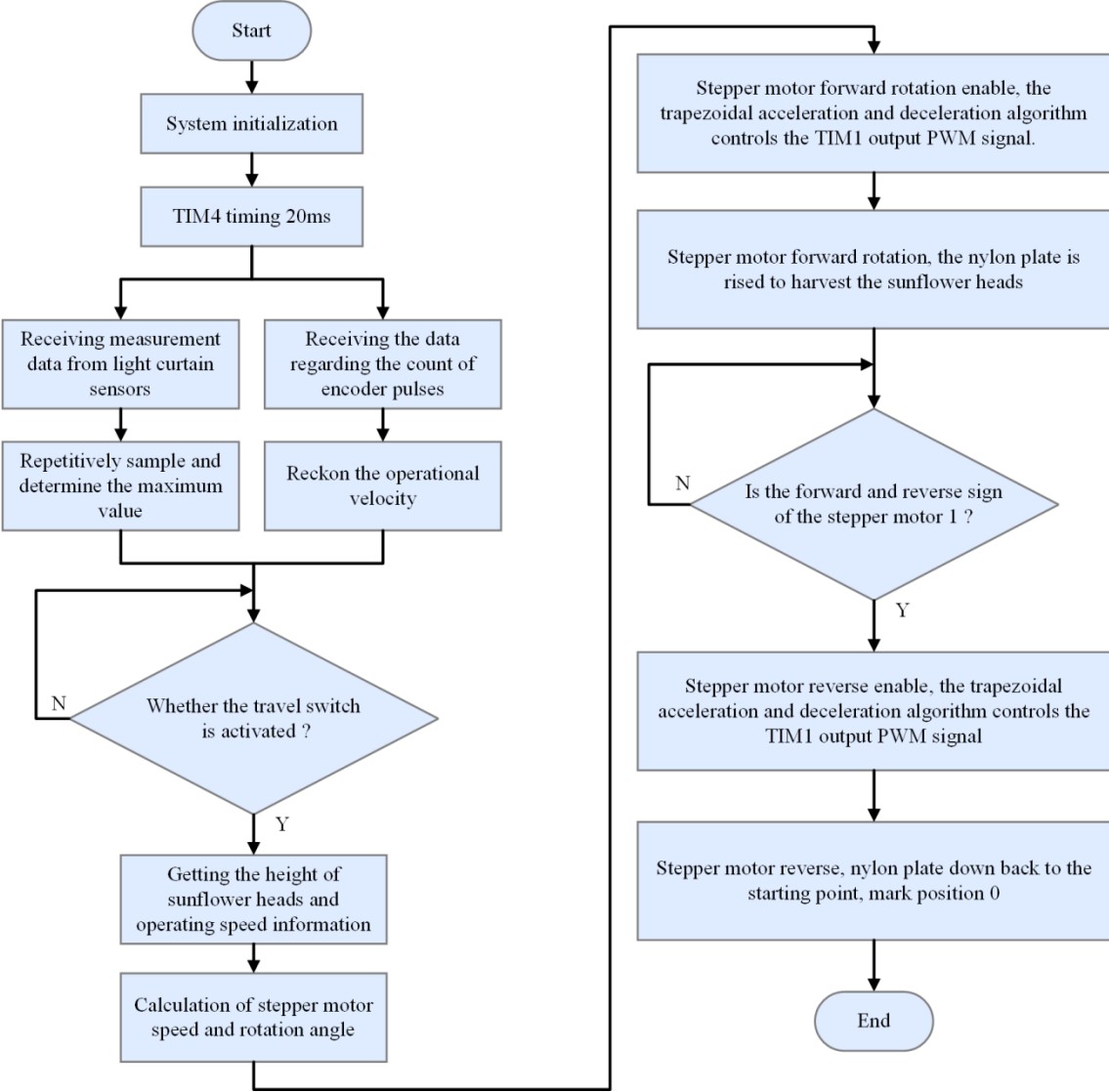

**Figure 6.** The software (Keil uvision5 MDKv518) flow diagram of the sunflower plate taking control system.

## 2.6. Design of Trapezoidal Acceleration and Deceleration Control Algorithm

The essence of the sunflower plate taking control system is the control of the stepping motor. According to the operation requirements, the nylon plate has higher requirements for motion speed and displacement during the rising and falling process. Therefore, there are strict requirements for the speed and angle control of the stepper motor. In this paper, a trapezoidal acceleration and deceleration control algorithm [21–23] is designed to control

the speed and rotation angle of the stepper motor. The algorithm is mainly composed of three stages: uniform acceleration, uniform speed and uniform deceleration.

At the beginning, the initial speed of the stepper motor is zero, with a certain uniform acceleration to a certain speed, a uniform motion is maintained, and then at a certain speed uniform deceleration to zero occurs. The uniform acceleration is equal to the uniform deceleration, which is a fixed value in the whole process of taking the sunflower plate. According to the maximum operating speed and the maximum operating height, the minimum acceleration and deceleration of the stepper motor are calculated to be 163.16 rad/s$^2$. In order to make the lifting process of the stepper motor more stable, 164 rad/s$^2$ is taken as the acceleration and deceleration rate in the trapezoidal acceleration and deceleration control algorithm. In order to control the stepper motor speed required for the stepper motor to reach Equation (7), the maximum speed of the stepper motor is greater than the motor speed calculated by Equation (7). The stepper motor is controlled to reach the required angle in Equation (8). The relationships between the maximum speed of the stepper motor, the rotation angle, the uniform acceleration time, the uniform speed time and the uniform deceleration time are shown in Equations (9)–(12):

$$T_a + T_b + T_c = T \tag{9}$$

$$T = \frac{l}{2v} - \Delta t \tag{10}$$

$$\frac{(T_a + T_c)n_{\max}}{2} + T_b n_{\max} = \theta \tag{11}$$

$$\beta = \frac{n_{\max}}{T_a} \tag{12}$$

In Equations (9)–(12), $T_a$ is the time of the uniform acceleration phase (s), $T_b$ is the time of the uniform speed stage (s), $T_c$ is the time of the uniform deceleration phase (s), $T$ is the rising and falling time of a single sunflower plate harvesting operation (s), $n_{\max}$ is the maximum speed of the stepper motor during acceleration and deceleration (rad/s), $\theta$ is the angle of rise and fall of a single sunflower harvesting operation (rad), $\beta$ is the rate of uniform acceleration and uniform deceleration (rad/s$^2$).

From Equations (9)–(12), the maximum speed of the stepper motor during acceleration and deceleration and the time of uniform acceleration, uniform speed and uniform deceleration of the stepper motor can be calculated for each sunflower harvesting operation.

The implementation process of the control algorithm is as follows. Suppose that $a$ is the step angle of the stepper motor, $\theta$ is the total angular displacement of the motor during the whole running time and $\theta_n$ is the angular displacement after the given nth pulse. $S$ is the total number of steps in the whole operation process of the stepper motor, $S_a$ is the number of steps in the acceleration stage of the stepper motor, $S_b$ is the number of steps in the uniform speed stage of the stepper motor and $S_c$ is the number of steps in the deceleration stage of the stepper motor. $t_n$ is the time when the nth pulse ends, and $T_n$ is the period of the nth pulse.

Starting the acceleration process in the stepper motor, Equation (13) of angular acceleration and angular displacement at this stage is obtained.

$$\theta_n = \frac{1}{2}\beta t_n^2 = n\alpha \tag{13}$$

$t_n$ is derived from Equation (13).

$$t_n = \sqrt{\frac{2n\alpha}{\beta}} \tag{14}$$

In summary, the calculation Equation (15) of the pulse period of the nth stepping motor can be obtained.

$$T_n = t_n - t_{n-1} = \sqrt{\frac{2\alpha}{\beta}}\left(\sqrt{n} - \sqrt{n-1}\right) \tag{15}$$

where the value of $n$ is 1, 2, . . . $S_a$.

From Equation (13), we can calculate the time $t_n$ at the end of the nth pulse transmission and the time $t_{n-1}$ at the end of the $n-1$ pulse transmission.

In the uniform speed process, it can be seen that the uniform speed running time is $T_b$ and the speed is $n_{\max}$. Then, the total angular displacement $\theta$ in the uniform stage can be calculated by Equation (16).

$$n_{\max}T_b = \theta \tag{16}$$

The time $t_n$ at the end of the nth pulse transmission can be calculated by Equation (17).

$$t_n = \frac{nT_b}{s_b} \tag{17}$$

Therefore, the period $T_n$ of the nth pulse of the uniform process can be calculated by Equation (18).

$$T_n = t_n - t_{n-1} = \frac{T_b}{s_b} \tag{18}$$

where the value of $n$ is 1, 2, . . . $S_b$.

The deceleration process and the acceleration process of the stepper motor are symmetrical, so the speed change laws of the deceleration process and the acceleration process are the same. The period $T_n$ of the nth pulse in the deceleration process of the stepper motor can be obtained from Equation (19).

$$T_n = t_n - t_{n-1} = T_c\left(\sqrt{\frac{s-n+1}{s_c}} - \sqrt{\frac{s-n}{s_c}}\right) \tag{19}$$

where the value of $n$ is 1, 2, . . . $S_c$.

In this paper, the control pulse of the stepper motor is generated by the TIM1 interruption of the controller. The TIM1 interruption outputs a drive pulse signal once. The interruption period is changed by continuously modifying the counter value of the timer, thereby changing the drive pulse frequency, so that the drive pulse period meets the above requirements and realizes the trapezoidal acceleration and deceleration motion control of the stepper motor.

### 2.7. Bench Experiment Materials and Conditions

The tools and instruments required for this experiment are: sunflower harvesting experiment bench (shown in Figure 7), tape measure (length 3 m), tape measure (length 50 m), high-precision electronic digital display Vernier caliper (accuracy 0.01 mm), Youli TSC precision electronic balance (range 0–15 kg, accuracy 0.01 g), scientific calculator, tray, etc.

The experimental materials were collected from the planting area of edible sunflower in Urumqi County, Xinjiang. The variety of edible sunflower was Sanrui 39, and the sampling date was 22 September 2022.

When the samples of the edible sunflower plates were collected, the sunflower plates were inserted into the plate for drying for 7 days, and the average moisture content of the sunflower plates was 36.2%. The experiment site was the Key Laboratory of Mechanical Equipment Institute of Xinjiang Academy of Agricultural Reclamation Sciences.

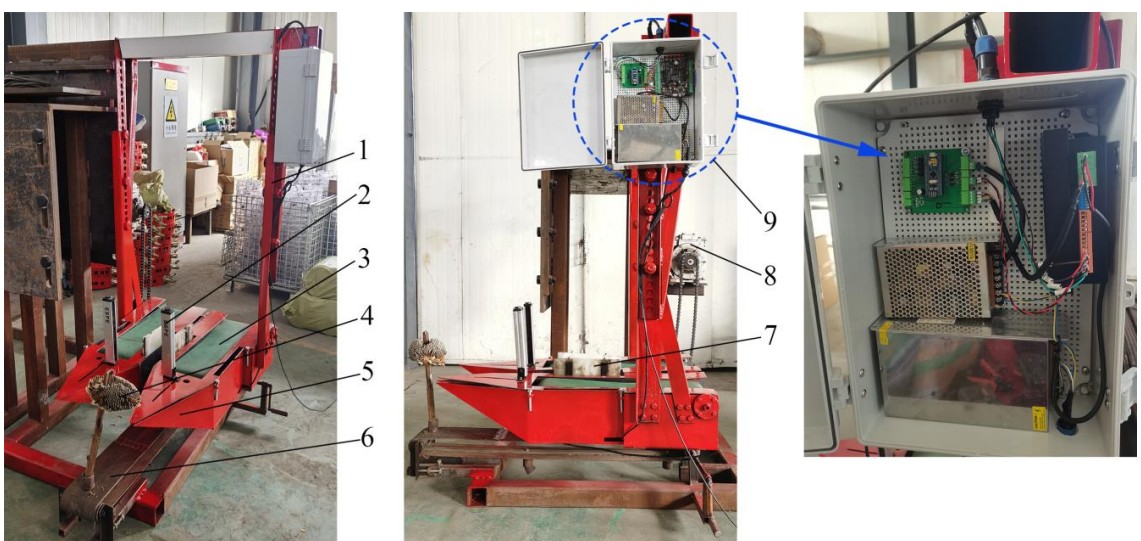

**Figure 7.** Sunflower harvest experiment bench. 1. Frame. 2. Light curtain sensor. 3. Conveyor belt. 4. Edible sunflower. 5. Divider. 6. Sunflower working belt. 7. Actuating mechanism of plate taking. 8. Operating motor. 9. Control box.

### 2.8. Field Experiment Materials and Conditions

The field experiment was carried out in the sunflower planting area of Urumqi County, Xinjiang from 15 to 20 September 2023, and the specific diagram of the experimental harvesting platform is shown in Figure 8. The edible sunflower variety is Sanrui 39, and the sunflower plates were inserted into the plate for drying for 7 days. The position of the sunflower plate placed on its rod is 650~800 mm from the surface height. The moisture content of this variety is between 31% and 35% at harvest. The experimental field requires that the edible sunflower plants have no lodging, and the row spacing and plant spacing are uniform.

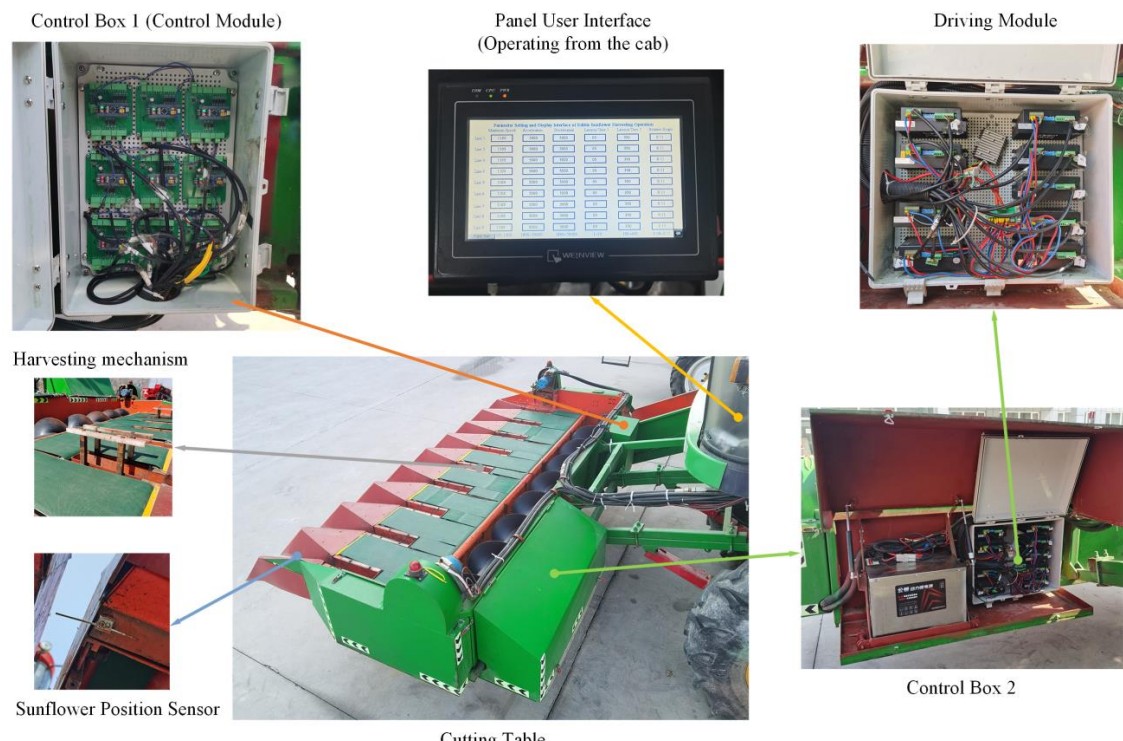

**Figure 8.** The installation diagram of each equipment on the harvesting platform.

### 3. Result and Analysis

*3.1. Bench Experiment*

3.1.1. Height Recognition Experiment of Sunflower Plate Placed on Its Rod

In order to ensure that the height measurement accuracy of sunflower plate placed on its rod meets the needs of field operation, the height identification experiment of sunflower plates was carried out. At present, the forward speed range of the sunflower harvester is 0.4~0.8 m/s [5,6]. In order to meet the actual operation requirements, the speed of the sunflower operation belt was set to the maximum forward speed of 0.8 m/s in the field. Twenty edible sunflower plants with different heights were selected and placed on the edible sunflower harvesting experiment bench for height measurement.

During the measurement, each plant was measured times, and the average value of the five measurements was used as the measured value of the light curtain sensor at the height of the sunflower plate placed on its rod (ESCL4805 measuring light curtain, optical axis interval of 5 mm). The actual height is measured by steel tape. Absolute error and relative error are used as the evaluation indexes of this experiment [24,25]. The absolute error in this experiment refers to the absolute value of the difference between the measured value of the light curtain sensor and the height of the actual sunflower plate placed on its rod. The relative error refers to the percentage of the absolute error to the actual height of the sunflower plate placed on its rod. Its calculation equations are as follows:

$$\Delta = |x - a| \tag{20}$$

$$E = \frac{|x - a|}{a} \times 100\% \tag{21}$$

where $\Delta$ is the absolute error (mm), $x$ is the measured value of the light curtain sensor (mm), $a$ is the height of the actual sunflower plate placed on its rod (mm), E is the relative error.

According to the above experiment methods, multiple experiments were carried out, and the experiment results are shown in Table 1 and Figure 9.

**Table 1.** Height measurement experiment results of sunflower disc placed on its rod.

| Number | Actual Height Value (mm) | The Measured Value of the Light Curtain Sensor (mm) | Absolute Error (mm) | Relative Error (%) |
|---|---|---|---|---|
| 1 | 53 | 50 | 3 | 5.66 |
| 2 | 64 | 60 | 4 | 6.25 |
| 3 | 76 | 75 | 1 | 1.32 |
| 4 | 79 | 75 | 4 | 5.06 |
| 5 | 86 | 85 | 1 | 1.16 |
| 6 | 98 | 95 | 3 | 3.06 |
| 7 | 108 | 105 | 3 | 2.78 |
| 8 | 111 | 110 | 1 | 0.90 |
| 9 | 117 | 115 | 2 | 1.71 |
| 10 | 123 | 120 | 3 | 2.44 |
| 11 | 127 | 125 | 2 | 1.57 |
| 12 | 132 | 130 | 2 | 1.52 |
| 13 | 134 | 130 | 4 | 2.99 |
| 14 | 137 | 135 | 2 | 1.46 |
| 15 | 143 | 140 | 3 | 2.10 |
| 16 | 157 | 155 | 2 | 1.27 |
| 17 | 168 | 165 | 3 | 1.79 |
| 18 | 177 | 175 | 2 | 1.13 |
| 19 | 188 | 185 | 3 | 1.60 |
| 20 | 192 | 190 | 2 | 1.04 |

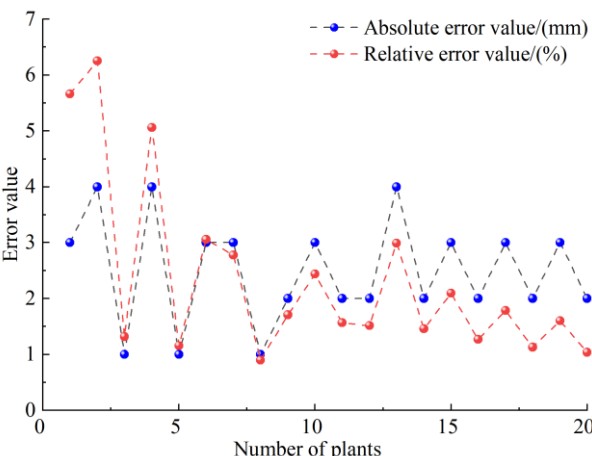

**Figure 9.** Experimental error line diagram of the height measurement accuracy of sunflower plate.

It can be seen from Table 1 and Figure 9 that the absolute error of the height measurement of each edible sunflower plate placed on its rod was less than 4 mm, the relative error range was 0.9~6.25% and the average relative error was less than 2.34%. It shows that the height measurement stability was good and the measurement error was small. The height measurement accuracy of the take-up of the sunflower plate control system meets the actual operation requirements.

### 3.1.2. The Speed Control Experiment of Taking Sunflower Plate

In order to ensure that the speed control effect of the nylon plate taking operation meets the operation requirements, the taking speed control experiment is carried out. The speed of the sunflower operating belt is still set to the maximum forward speed of the unit in the field of 0.8 m/s. According to the field investigation of the planting area in Xinjiang and the relevant literature, sunflower plates are usually placed on their rods at heights ranging from 650 to 800 mm above the surface [7,8]. It can be seen that the height difference of the sunflower plate placed on its rod manually is 150 mm. The relative height was selected as the experiment factor. The relative height is the distance between the top of the sunflower plate and the nylon plate at the starting point. The maximum thickness of the sunflower plate is 50 mm. To sum up, in order to ensure that the plate taking part can complete the sunflower plate taking operation, the minimum relative height of the sunflower plate placed on its rod is 50 mm, and the maximum is 200 mm. The three levels of relative height in the experiment are set to 50 mm, 125 mm and 200 mm. The stepper motor driver fraction was set to 32, by setting different levels in the program combination experiment. The actual speed of the nylon plate was calculated by the encoder of the stepping motor, and the theoretical speed of the nylon plate was calculated by Equation (22). The speed and time curve of the nylon plate was drawn, as shown in Figure 10.

$$v = \begin{cases} \frac{\beta t d}{2i}, 0 \leq t < T_a \\ \frac{n_{max} d}{2i}, T_a \leq t < T_b \\ \frac{(n - \beta t) d}{2i}, T_b \leq t \leq T_c \end{cases} \tag{22}$$

It can be seen from Figure 9 that the actual velocity curve of the nylon plate basically conforms to the law of trapezoidal acceleration and deceleration motion. The actual speed of the nylon plate can change with the change in the working speed and the height of the sunflower plate placed on the sunflower rod. The speed error is less than 0.028 m/s, and the control effect is good, which meets the design requirements.

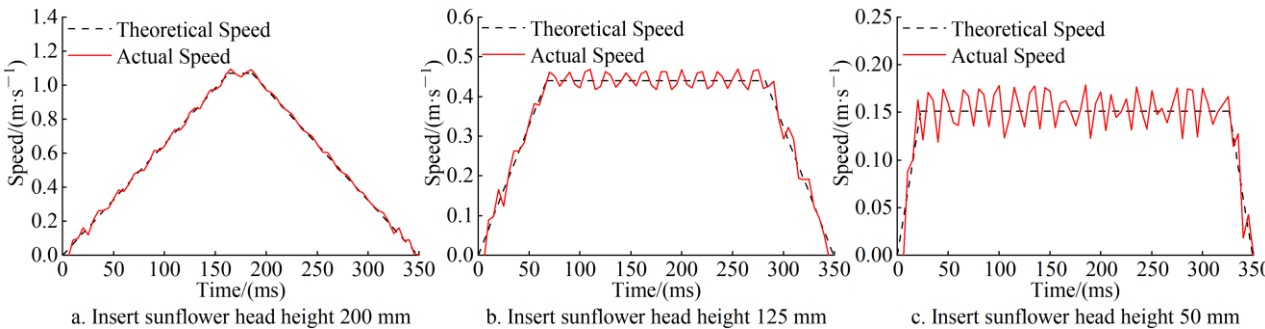

a. Insert sunflower head height 200 mm   b. Insert sunflower head height 125 mm   c. Insert sunflower head height 50 mm

**Figure 10.** The fitting curve of the theoretical speed and the actual speed of the harvest sunflower plate components.

### 3.1.3. Experiment of Picking Sunflower Plate

In order to ensure that the plate taking control system can control the sunflower harvesting table to complete the plate harvesting operation under the field operating conditions, in this experiment, the forward speed and relative height (the distance between the top of the sunflower plate and the conveyor belt plane of the sunflower harvesting table) were selected as the experimental factors. In order to simulate the field conditions, the forward speed is set to 0.40 m/s, 0.60 m/s and 0.80 m/s. The relative height of the sunflower plate placed on its rod is set to 50 mm, 125 mm and 200 mm. The fine fraction of the stepper motor driver is set to 32, and the experiment is performed by setting each level combination in the program. In order to comprehensively evaluate the operation quality and performance of the sunflower harvesting table and the plate taking control system, referring to DB65/T3541-2013 'Technical Protocols of production mechanization in Sunflower' [26], the absolute error of the plate taking displacement, the relative error of the plate taking displacement and the total loss rate during the plate taking process were selected as the experiment indexes [27–30]. The theoretical displacement of the nylon plate is the relative height. The actual displacement of the plate is the actual rising height of the nylon plate measured by the tape measure. The experimental factors and levels are shown in Table 2.

**Table 2.** Factors and levels of experiment.

| Coded Value | Coding Factor | |
| --- | --- | --- |
| | Forward Speed (m·s$^{-1}$) | Relative Height of Harvest (mm) |
| 1 | 0.4 | 50 |
| 2 | 0.6 | 125 |
| 3 | 0.8 | 200 |

The absolute error in this experiment refers to the absolute value of the difference between the theoretical displacement and the actual displacement of the plate taking. The relative error refers to the percentage of the absolute error of the experiment to the theoretical displacement. The equation for calculating the total loss rate during plate taking is as follows:

$$S_1 = \frac{W_a}{W_a + W_b} \times 100\% \tag{23}$$

where $S_1$ is the total loss rate of the sunflower plate (%), $W_a$ is the loss of grain mass in the experiment area (kg), $W_b$ is the grain quality harvested in the experiment area (kg).

The plate taking experiment was carried out with reference to GB/T 8097-2008 'Equipment for harvesting-Combine harvesters-Test procedure' [31]. According to the experimental scheme, 13 test points were selected for a response surface analysis test. The experiment area is divided into two parts, the stable area and the experiment area. The stable area is 2.4 m long and the experiment area is 6 m long. The stable area is simulated by the

operation motor driving the operation belt to rotate two circles, and the experiment area is simulated by the operation motor driving the operation belt to rotate five circles. Before each experiment, the scattered sunflower plates, broken plants and naturally fallen seeds in the experiment area were cleaned. Each group of experiments was repeated five times, the arithmetic mean value of the five experiments was taken as the experiment result at this level, and the data were recorded. The experimental design and results are shown in Table 3.

**Table 3.** Experiment scheme and results of plate taking.

| Number | Operating Speed (m/s) | Theoretical Displacement (mm) | Actual Displacement (mm) | Absolute Error (mm) | Relative Error (%) | Total Loss Rate (%) |
|---|---|---|---|---|---|---|
| 1 | 0.4 | 50 | 50.60 | 0.60 | 1.20 | 1.31 |
| 2 | 0.4 | 125 | 124.20 | 0.80 | 0.64 | 1.62 |
| 3 | 0.4 | 200 | 198.80 | 1.20 | 0.60 | 1.98 |
| 4 | 0.6 | 50 | 50.80 | 0.80 | 1.60 | 1.43 |
| 5 | 0.6 | 125 | 126.40 | 1.40 | 0.87 | 1.74 |
| 6 | 0.6 | 200 | 201.25 | 1.25 | 1.12 | 2.11 |
| 7 | 0.8 | 50 | 50.40 | 0.40 | 0.80 | 1.55 |
| 8 | 0.8 | 125 | 126.20 | 1.20 | 0.96 | 1.83 |
| 9 | 0.8 | 200 | 201.40 | 1.40 | 0.70 | 2.26 |
| 10 | 0.6 | 125 | 125.40 | 0.40 | 0.32 | 1.85 |
| 11 | 0.6 | 125 | 126.20 | 1.20 | 0.96 | 1.87 |
| 12 | 0.6 | 125 | 124.40 | 0.60 | 0.48 | 1.91 |
| 13 | 0.6 | 125 | 126.20 | 1.20 | 0.96 | 1.82 |

Design-Expert 10.0.3 data analysis software was used to fit the experimental data by multiple regression. The forward speed of the machine and the relative height of the harvest are A and B, respectively. The total loss rate was used as the response value for multiple regression fitting. The quadratic polynomial regression model equation (as shown in Equation (24)) is obtained. The regression model coefficients and significance test results of analysis of variance are shown in Table 4.

$$Y = 1.85 + 0.13A + 0.24B + 0.06AB - 0.095A^2 - 0.05B^2 \tag{24}$$

**Table 4.** Analysis of variance of the regression model.

| Source of Variance | Total Loss Rate | | | |
|---|---|---|---|---|
| | Sum of Squares | Degrees of Freedom | F | p |
| Model | 0.52 | 5 | 68.77 | <0.0001 *** |
| A | 0.099 | 1 | 65.61 | <0.0001 *** |
| B | 0.36 | 1 | 235.89 | <0.0001 *** |
| AB | 0.014 | 1 | 9.56 | 0.0175 * |
| $A^2$ | 0.025 | 1 | 16.67 | 0.0047 ** |
| $B^2$ | $7.00 \times 10^{-3}$ | 1 | 4.65 | 0.068 |
| Residual | 0.11 | 7 | | |
| Lack-of-fit | $5.86 \times 10^{-3}$ | 3 | 1.67 | 0.3092 |
| Error | $4.68 \times 10^{-3}$ | 4 | | |
| Sum | 0.53 | 12 | | |

Note: * indicates significance ($p < 0.05$); ** indicates high significance ($p < 0.01$); *** indicates high significance ($p < 0.001$).

Further regression analysis was performed on the model and regression coefficients. The results are shown in Table 4, where the *F* value is an important indicator to evaluate the influence of each variable on the response value. The larger the *F* value, the higher the contribution of the model components to the response. When the significance test

probability $p < 0.05$, it is revealed that the variable has a significant effect on the response value, which is of mathematical statistical significance. It can be seen from Table 4 that the regression model $p < 0.001$ (extremely significant), indicating that the model has a good degree of fitting, and the corresponding regression values of the regression equation can be predicted, indicating that the equation is highly reliable. By analyzing the variance data of the loss rate, it can be seen that the forward speed of the first item and the relative height of the harvest have a very significant impact on the total loss rate ($p < 0.01$). The main effect of each factor is analyzed as follows: forward speed > relative height of harvest. The quadratic interaction AB had a significant effect on the total loss rate ($p < 0.05$).

The effect of the interaction between the forward speed of the machine and the relative height of the harvest on the total loss rate is shown in Figure 11. It can be seen from the diagram that in the AB interaction surface, the change slope of the total loss rate increases linearly with the increase in the relative height of the harvest, and the change slope of the loss rate increases slowly with the increase in the forward speed. When the relative height of the harvest is different, the change slope of the loss rate is randomly different from the increase in the forward speed. Therefore, the interaction between the relative height of the harvest and the forward speed of the machine on the loss rate is significant, and the relative height of the harvest is greater than the forward speed of the machine. The analysis of the reason is as follows. With the increase in the relative height of the harvest, the success rate of taking sunflower plates is reduced. When the relative height increases, the time for taking the sunflower plate parts will increase. It will also increase the contact time between the sunflower plate and the harvester. At this time, the harvester running at a fixed forward speed will also cause a certain impact on sunflower plants, which will greatly increase the total loss rate of grains. When the forward speed increases in the range of 0.4~0.6 m/s, some sunflower plates can still enter the cutting table due to inertia. When the forward speed increases in the range of 0.6~0.8 m/s, some sunflower plates collide with the cutting table and fall to the ground, thus increasing the total loss rate.

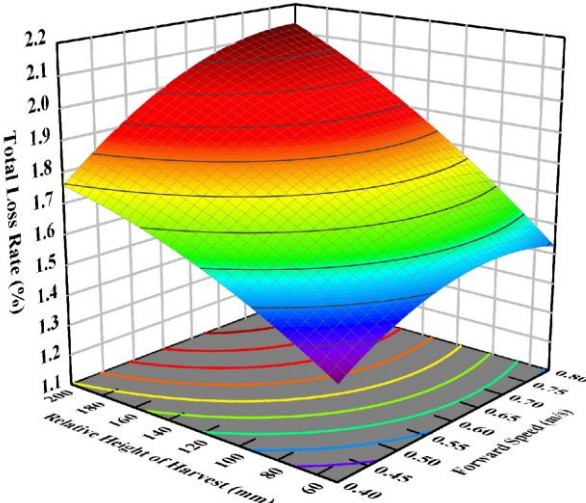

**Figure 11.** Effect of forward speed and relative height of harvest on total loss rate.

According to the results of the plate taking experiment, the maximum positioning error of the sunflower harvesting experiment bench is 1.25 mm, the maximum relative error is 1.2%, the average relative error is 0.94% and the overall grain loss rate is not more than 2.26% under the field operating conditions. It shows that the system has high positioning accuracy and small grain loss rate, which meets the requirements of sunflower harvesting operation under the segmented harvesting mode.

### 3.2. Field Experiment

In order to verify the performance and effectiveness of the plate taking control system, it is ensured that the plate taking control system can meet the requirements of the harvester to complete the plate taking operation within the field operation speed range. The field verification experiment was carried out on the sunflower harvester equipped with the plate taking control system, as shown in Figure 12. This experiment was based on GB/T8097-2008 'Equipment for harvesting-Combine harvesters-Test procedure' [31]. The experiment field selected for the experiment consists of three parts: the stable area, the measurement area and the parking area. The length of the experiment area is 25 m, the stable area and the parking area are 20 m. Before the harvesting operation of edible sunflower, it is necessary to remove the scattered sunflower plates, broken plants and naturally fallen seeds in the measurement area. During the experiment, it is necessary to repeat the experiment and record the relevant experiment data to ensure the accuracy of the experiment results. The reliability evaluation of the plate picking operation of the edible sunflower harvesting table is mainly based on JB/T6287-2008 'Reliability determination test methods for grain combine harvesters' [32]. The quality requirements of field operations refer to DB65/T3541-2013 'Technical Protocols of production mechanization in Sunflower' [26]. The total loss rate of harvest was selected as the evaluation index. The experiment results take the mean of multiple sets of experiments. The equation for calculating the total loss rate during plate taking is as follows:

$$S_2 = \frac{W_d}{W_s + W_d} \times 100\% \tag{25}$$

where $S_2$ is the total loss rate of sunflower plates (%), $W_d$ is the mass of the sunflower plates lost in the measurement area (including the fallen grain, the fallen sunflower plates and the unpicked sunflower plates) (kg), $W_s$ is the weight of the sunflower plates harvested in the measurement area (kg).

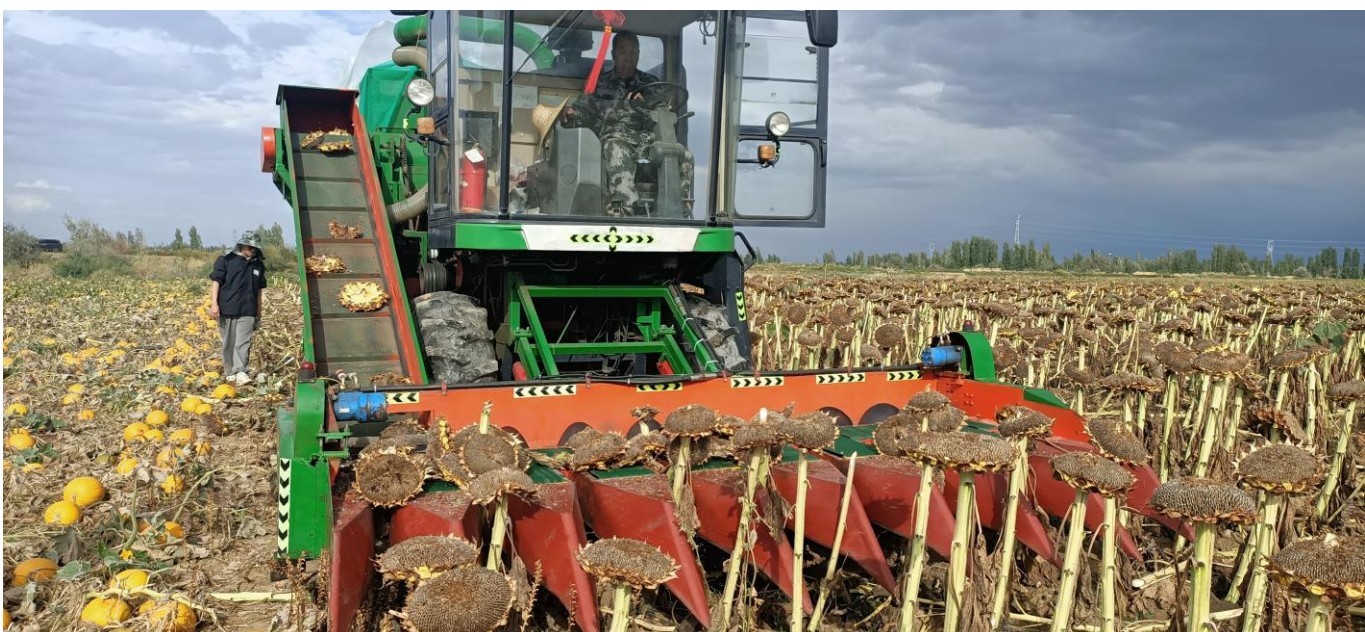

**Figure 12.** Sunflower harvest field experiment.

For the sunflower harvester equipped with the plate taking control system, the field verification experiment of the sunflower plate taking was carried out. The forward speed of the unit is set to slow second gear (0.4 m/s), slow third gear (0.6 m/s) and slow fourth gear (0.8 m/s). During the field operation, each group of experiments was repeated three times, and then the arithmetic mean of the total harvest loss rate was taken as the experiment result. The results are shown in Table 5.

**Table 5.** Scheme and results of field experiment.

| Speed of Advance (m/s) | Total Loss Rate (%) |
|---|---|
| 0.4 | 2.87 |
| 0.6 | 2.32 |
| 0.8 | 3.16 |

It can be seen from Table 5 that when the forward speed of the harvester was 0.4~0.8 m/s, the total loss rate of the harvest was less than 5%. When the forward speed was 0.6 m/s, the minimum harvest loss rate of the unit was 2.32%. It shows that the control system can better complete the plate picking operation, the system reliability is strong and it can meet the requirements of sunflower harvesting operation.

## 4. Discussion and Perspectives

### 4.1. Discussion on Key Parameters of Harvester

In this study, the effects of forward speed and relative height of harvest on total grain loss rate were proved by experiments. The relative height determines the time of a single harvest operation. The forward speed of the harvester determines the working area per unit time. The relationship between the two has a key impact on the high-quality harvest of edible sunflowers. By optimizing these two parameters, the total loss rate of edible sunflower seeds can be effectively reduced, and the harvest efficiency and economic benefits of crops can be improved.

### 4.2. Comparison with Related Research Results

By comparing with other related research [6–9], we found that our research results have the same place in the range of test factors, test evaluation index and so on, but our research focus is to systematically evaluate the comprehensive performance of the tray control system of the sunflower harvesting table by means of a bench test and field verification test. Taken together, these studies have revealed the development trend of edible sunflower harvesting machinery, which is of great significance for further improving the mechanized harvesting efficiency of edible sunflower.

### 4.3. Limitations of This Study

First of all, the specific conditions of the experimental environment may have an impact on the results of edible sunflower harvest, which needs to be verified in a wider range of scenarios in the future. Secondly, it should be noted that the sample size of this study is relatively small and may not fully represent the entire target group. Finally, our research methods may have some limitations, for example, we may not be able to control all potential interference variables. Therefore, in future studies, we recommend expanding the sample size and considering using more advanced research methods to further validate our results.

### 4.4. Future Project Implementation Plan

(1) Future research can further explore other factors that may affect the results, in-depth study of the interaction between the various factors, more comprehensive study to improve the performance of the control system.

(2) We recommend conducting more extensive research on different target groups, including different varieties of edible sunflowers or harvests under different geographical conditions, to understand whether the conclusions of this study apply to a wider range of situations.

(3) Long-term experiments could be carried out to evaluate the durability and stability of the system in actual operation.

(4) It is recommended to establish cooperation with experts in related fields such as machine vision and crop phenotype and further improve the reliability of the control

system by using advanced technical means such as multi-sensor fusion and jointly explore new development directions.

## 5. Conclusions

In order to meet the needs of segmented harvesting of edible sunflower in Xinjiang, this paper designs a plate taking control system of an edible sunflower harvesting table based on the principle of manual plate taking. The system can replace the manual completion of the plate taking operation and can effectively reduce the grain loss rate during the harvest of edible sunflower. The main conclusions of this investigation are as follows:

(1) According to the analysis of the operating parameters of the sunflower plate and the working principle of the harvesting device, the overall scheme of the plate control system was determined. The system is composed of a working speed detection sensor, position detection sensor, height detection sensor, controller, driver and stepper motor. According to the relevant parameters, the relationships between the speed of the stepper motor, the rotation angle, the working speed and the height of the sunflower plate placed on the sunflower rod are established.

(2) The sunflower harvest experiment bench was built to carry out the bench experiment of the control system. The results showed that the absolute error of the height measurement of each edible sunflower was less than 4 mm. The actual speed curve of the plate taking components basically conforms to the trapezoidal acceleration and deceleration motion law, and the speed error of the plate taking components does not exceed 0.028 m/s. Under the simulated field operating conditions, the maximum positioning error was 1.25 mm, the maximum relative error was 1.2%, the average relative error was 0.94%, and the overall grain loss rate was less than 2.26%. It shows that the algorithm of the system is reliable, the positioning accuracy is high, the grain loss rate is small and the picking operation can be completed well.

(3) The plate taking control system designed in this paper was installed on the sunflower harvester, and the field experiment was carried out. The results showed that the total loss rate was less than 5% when the forward speed of the harvester was 0.4~0.8 m/s. When the forward speed is 0.6 m/s, the minimum harvest loss rate of the unit is 2.32%. It shows that the control system meets the requirements of the harvesting operation of the edible sunflower in the segmented harvesting mode, which improves the harvesting efficiency of the edible sunflower and reduces the harvesting loss rate.

**Author Contributions:** Conceptualization, F.P., B.L. and C.J.; methodology, F.P. and J.C.; software, F.P., J.C. and L.H.; validation, F.P., C.J. and H.Z.; formal analysis, F.P. and Y.D.; investigation, F.P., L.H. and H.Z.; resources, F.P. and C.J.; data curation, F.P. and L.H.; writing—original draft preparation, F.P.; writing—review and editing, F.P. and C.J.; visualization, F.P. and H.Z.; supervision, C.J.; project administration, F.P., B.L. and C.J.; funding acquisition, B.L. and C.J. All authors have read and agreed to the published version of the manuscript.

**Funding:** This research was funded by the Key Scientific and Technological Projects in Key Areas of the Xinjiang Production and Construction Corps (No. 2021AB001).

**Institutional Review Board Statement:** Not applicable.

**Data Availability Statement:** Data are contained within the article.

**Conflicts of Interest:** The authors declare no conflicts of interest.

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
