# Peer review of "Design and Experiment of Plate Taking Control System of Edible Sunflower (Edulis helianthus catino L.) Harvester"

_agriculture, doi:10.3390/agriculture14040592_

Round 1
Reviewer 1 Report
Comments and Suggestions for Authors
The manuscript presents the design of a plate-taking control system for an edible sunflower harvesting table based on the principle of manual plate-taking, which is interesting. Conducting research of this nature is crucial for effectively reducing the grain loss rate during the harvest of edible sunflowers. However, I have identified some issues that need to be addressed and corrected.
1. It would be beneficial to include the rationale for selecting the specific input parameters, namely forward speed and the relative height (the distance between the top of the sunflower plate and the conveyor belt plane of the sunflower harvesting table). Why were other design parameters not taken into consideration?
2. Why did you choose 5 replications for each group of experiments? On what basis did you determine this quantity, and why did you not consider opting for 6 or more repetitions?
3. Incorporate a table presenting the Analysis of Variance (ANOVA) for assessing the impact of factors on response indicators. Additionally, discuss the significant factors and their interactions in the main text.
4. It would be beneficial to include the effects of each independent parameter—namely, forward speed and the relative height—on the dependent parameter, namely, total loss rate. Please consider adding separate subheadings under the 'Discussion' section to elaborate on these relationships.
5. Please improve the readability of the captions for Figures 9 and 10 by rewriting them.
6. It would be beneficial to include the limitations of this study and provide suggestions for future work in the conclusion section.
Reviewer 2 Report
Comments and Suggestions for Authors
The article is devoted to the development of an automated control system for plates for collecting sunflowers in order to minimize losses.
The article has scientific and practical interest, is written in a language that is understandable to the reader and is well structured.
For publication, I consider it necessary to correct the following comments:
1) What automation software is used? Needs to be described.
2) Provide a screenshot of the user interface from the panel or computer screen
3) Have the dynamics of the control object system and the control system been studied using methods of automatic control theory? Have dynamic models been built? Perhaps you have done this in other works, then you can refer to them
4) Provide a photo of the assembled control system (in Figure 7 the control unit is visible, but from a distance; it would be good to zoom in)
5) It is also advisable to provide a video of the operation of the control system, you can submit it as an additional file, or provide a link to this video
6) Where is the control unit located on the combine harvester? If everything is clear with bench tests, then there are questions regarding field tests. Please draw a diagram of the installation of sensors and motors and other equipment on the combine, following the example as shown in Fig. 7. Or is it still under development? Because in the photo (Fig. 8) you can only see the combine in operation, but where is your automation? If field tests with a new development are just being planned, then write about it in the future, since the subsection with field studies and its results look weak.
7) You also need to add a “Discussion and Perspectives” section, where you compare your experimental results with the results of similar studies by other authors. And also write how you plan to develop your project
Reviewer 3 Report
Comments and Suggestions for Authors
When specifying the object of research in the title of a manuscript, such as a plant (here an edible sunflower) or an animal, it is necessary to provide its Latin name.
The very idea of harvesting whole sunflower inflorescences is questionable. Today it is possible to design and build a machine to harvest anything and perform computer simulations to evaluate it for agricultural practice. The main purpose of growing edible sunflower is to obtain the seeds themselves, e.g. using a combine harvester with an appropriate attachment for oil for humans and possibly oil cake for animals. The inflorescences themselves are collected only occasionally and individually for e.g. aesthetic purposes. What is the purpose of such a collection in this work?
The entry of equation/inequality 1 and 3 is questionable.
no specifics
Round 2
Reviewer 1 Report
Comments and Suggestions for Authors
This revised version is significantly better than before. All comments have been considered, so the paper can be accepted in this revised version.
Reviewer 2 Report
Comments and Suggestions for Authors
The new version of the article makes up for the shortcomings that were in the first version. In particular, the authors described the software, provided screenshots from the panel screen, improved the quality of the drawings, provided a video of the control system in action, improved the section on field experiments by adding analysis of variance, and added a section “Discussion and Perspectives.”
In this form, in my opinion, the article can already be published. I recommend that the videos provided by the authors be included in the article as additional files or as a link in the article (at the discretion of the editors). I also recommend adding a video from field tests (or at least a photo, in addition to Figure 12, so that the operation of the control system can be seen).
